# CHARACTERIZING AND MEASURING THE SIMILARITY OF NEURAL NETWORKS WITH PERSISTENT HOMOLOGY

## ABSTRACT

Characterizing the structural properties of neural networks is crucial yet poorly understood, and there are no well-established similarity measures between networks. In this work, we observe that neural networks can be represented as abstract simplicial complex and analyzed using their topological 'fingerprints' via Persistent Homology (PH). We then describe a PH-based representation proposed for characterizing and measuring similarity of neural networks. We empirically show the effectiveness of this representation as a descriptor of different architectures in several datasets. This approach based on Topological Data Analysis is a step towards better understanding neural networks and a useful similarity measure.

## 1 INTRODUCTION

Machine learning practitioners can train different neural networks for the same task. Even for the same neural architecture, there are many hyperparameters, such as the number of neurons per layer or the number of layers. Moreover, the final weights for the same architecture and hyperparameters can vary depending on the initialization and the optimization process itself, which is stochastic. Thus, there is no direct way of comparing neural networks accounting for the fact that neural networks solving the same task should be measured as being similar, regardless of the specific weights. This also prevents one from finding and comparing modules inside neural networks (e.g., determining if a given sub-network does the same function as other sub-network in another model). Moreover, there are no well-known methods for effectively characterizing neural networks.

This work aims to characterize neural networks such that they can be measured to be similar once trained for the same task, with independence of the particular architecture, initialization, or optimization process. To characterize neural networks and measuring their similarity, we assume two different similarities that should be related if the characterization is correct, namely, 1. similarity between the tasks a model is trained on, and 2. similarity between neural networks themselves considered as mathematical objects.. We do not provide a mathematical definition for the first one, rather an intuitive one, and understand is as qualitative similarity between semantically close tasks. For instance, we think that MNIST (10 labels) is closer to MNIST restricted to 2 labels than to language identification, and that MNIST is closer to MNIST restricted to 8 labels than to MNIST restricted to 2 labels. For the second one, in this work we propose a similarity measure.

Ideally, this similarity measure should then capture 1. the semantic similarity between the underlying tasks neural networks have been trained on (understanding that networks performing similar tasks should be similar, regardless of their specific weights and architectures), and 2. intrinsic properties of the neural network should also be captured to some degree. Thus, we consider two scenarios, that is, comparing NNs trained for similar tasks, using the mentioned task similarity, and comparing neural networks trained on identical tasks but with different architectures.

We focus on Multi-Layer Perceptrons (MLPs) for the sake of simplicity. We start by observing that we can represent a neural network as a directed weighted graph to which we can associate certain topological concepts.[1] Considering it as a simplicial complex, we obtain its associated Persistent Diagram. Then, we can compute distances between Persistent Diagrams of different neural networks.

---

[1]See Jonsson (2007) for a complete reference on graph topology.

The proposed experiments aim to show that the selected structural feature, Persistent Homology, serves to relate neural networks trained for similar problems and that such a comparison can be performed by means of a predefined measure between the associated Persistent Homology diagrams. To test the hypothesis, we study different classical problems (MNIST, Fashion MNIST, CIFAR-10, and language identification and text classification datasets), different architectures (number and size of layers) as well as a control experiment (input order, to which neural network similarity should be invariant). The code and results are fully open source in the Supplementary Material under a MIT license.

In summary, the main contributions of this work are the following: 1. We propose an effective graph characterization strategy of neural networks based on Persistent Homology. 2. Based on this characterization, we suggest a similarity measure of neural networks. 3. We provide empirical evidence that this Persistent Homology framework captures valuable information from neural networks and that the proposed similarity measure is meaningful.

The remainder of this paper is organized as follows. In Section 2, we go through the related work. Then, in Section 3 we describe our proposal and the experimental framework to validate it. Finally, in sections 4 and 5 we report and discuss the results and arrive to conclusions, respectively.

## 2 RELATED WORK

One of the fundamental papers of Topological Data Analysis (TDA) is presented in Carlsson (2009) and suggests the use of Algebraic Topology to obtain qualitative information and deal with metrics for large amounts of data. For an extensive overview of simplicial topology on graphs, see Giblin (1977); Jonsson (2007). Aktas et al. (2019) provide a thorough analysis of PH methods.

More recently, a number of publications have dealt with the study of the capacity of neural networks using PH. Guss & Salakhutdinov (2018) characterize learnability of different neural architectures by computable measures of data complexity. Donier (2019) propose the concept of *spatial capacity allocation analysis*. Konuk & Smith (2019) propose an empirical study of how NNs handle changes in topological complexity of the input data.

Rieck et al. (2019b) introduce the *neural persistence* metric, a complexity measure based on TDA on weighted stratified graphs. This work suggests a representation of the neural network as a multipartite graph. They perform the filtering of the Persistent Homology diagrams independently for each layer. As the filtration contains at most 1-simplices (edges), they only capture zero-dimensional topological information, i.e. connectivity information. When consecutive layer analysis is used, the global topology of the network is not taken into account making the strong assumption that the NNs encode the learned information layer pairwise exclusively. Additionally, there are trivial global transformations of a NN that are not captured by analyzing pairs of layers:

- Superfluous cycle insertions: for example, add two neurons and connect their input to a single neuron in a lower layer and their two outputs to a single output neuron in an upper layer with opposite weights.
- Identity layer insertions: for instance, insert an intermediate identity layer with neurons and trivially connect to the next layer.
- Non-planar neural networks analysis: the analysis of neural networks that use multiple connections between non-consecutive layers require higher order topological analysis.

In terms of pure neural network analysis, there are relevant works, like Hofer et al. (2020), that study topological regularization. Clough et al. (2020) introduce a method for training neural networks for image segmentation with prior topology knowledge, specifically via Betti numbers. Corneanu et al. (2020) try to estimate (with limited success) the performance gap between training and testing via neuron activations and linear regression of the Betti numbers. This type of representation depend on the input data and not only on the NN function under study. Instead, we are interested in characterising and comparing NNs as functions, independently of the data to which they are applied.

On the other hand, topological analysis of decision boundaries has been a very prolific area. Ramamurthy et al. (2019) propose a labeled Vietoris-Rips complex to perform PH inference of decision boundaries for quantification of the complexity of neural networks. Naitzat et al. (2020) experiment

on the PH of a wide range of point cloud input datasets for a binary classification problems to see that NNs transform a topologically rich dataset (in terms of Betti numbers) into a topologically simpler one as it passes through the layers. They also verify that the reduction in Betti numbers is significantly faster for ReLU activations than hyperbolic tangent activations.

Regarding neural network representations, one of the most related works to ours, Gebhart et al. (2019), focuses on topological representations of neural networks. They introduce a method for computing PH over the graphical activation structure of neural networks, which provides access to the task-relevant substructures activated throughout the network for a given input.

Interestingly, in Watanabe & Yamana (2020), authors work on neural network representations through simplicial complexes based on deep Taylor decomposition and they calculate the PH of neural networks in this representation. In Chowdhury et al. (2019), they use directed homology to represent MLPs. They show that the path homology of these networks is non-trivial in higher dimensions and depends on the number and size of the network layers. They investigate homological differences between distinct neural network architectures.

As far as neural network similarity measures are concerned, the literature is not especially prolific. In Kornblith et al. (2019), authors examine similarity measures for representations (meaning, outputs of different layers) of neural networks based on canonical correlation analysis. However, note that this method *compares neural network representations (intermediate outputs), not the neural networks themselves*. Remarkably, in Ashmore & Gashler (2015), authors *do* deal with the intrinsic similarity of neural networks themselves based on Forward Bipartite Alignment. Specifically, they propose an algorithm for aligning the topological structures of two neural networks. Their algorithm finds optimal bipartite matches between the nodes of the two MLPs by solving the well-known graph cutting problem. The alignment enables applications such as visualizations or improving ensembles. However, the methods only works under very restrictive assumptions,[2] and this line of work does not appear to have been followed up.

Finally, we note that there has been a considerable growth of interest in applied topology in the recent years. This popularity increase and the development of new software libraries,[3] along with the growth of computational capabilities, have empowered new works. Some of the most remarkable libraries are Ripser Tralie et al. (2018); Bauer (2021), and Flagser Lütgehetmann et al. (2019). They are focused on the efficient computation of PH. For GPU-Accelerated computation of Vietoris-Rips PH, Ripser++ Zhang et al. (2020) offers an important speedup. The Python library we are using, Giotto-TDA Tauzin et al. (2020), makes use of both above libraries underneath.

We have seen that there is a trend towards the use of algebraic topology methods for having a better understanding of phenomena of neural networks and having more principled deep learning algorithms. Nevertheless, little to no works have proposed neural network characterizations or similarity measures based on intrinsic properties of the networks, which is what we intend to do.

## 3 METHODOLOGY

In this section, we propose our method, which is heavily based on concepts from algebraic topology. We refer the reader to the Supplementary Material for the mathematical definitions. In this section, we also describe the conducted experiments.

Intrinsically characterizing and comparing neural networks is a difficult, unsolved problem. First, the network should be represented in an object that captures as much information as possible and then it should be compared with a measure depending on the latent structure. Due to the stochasticity of both the initialization and training procedure, networks are parameterized differently. For the same task, different functions that effectively solve it can be obtained. Being able to compare the trained networks can be helpful to detect similar neural structures.

We want to obtain topological characterizations associated to neural networks trained on a given task. For doing so, we use the Persistence Homology (from now on, PH) of the graph associated to a

---

[2]For example, the two neural networks "must have the same number of units in each of their corresponding layers", and the match is done layer by layer.

[3]https://www.math.colostate.edu/~adams/advising/appliedTopologySoftware/

neural network. We compute the PH for various neural networks learned on different tasks. We then compare all the diagrams for each one of the task.

More specifically, for each of the studied tasks (image classification on MNIST, Fashion MNIST and CIFAR-10; language identification, and text classification on the Reuters dataset),[4] we proceed as follows: 1. We train several neural network models on the particular problem. 2. We create a directed graph from the weights of the trained neural networks (after changing the direction of the negative edges and normalising the weights of the edges). 3. We consider the directed graph as a simplicial complex and calculate its PH, using the weight of the edges as the filtering parameter, which range from 0 to 1. This way we obtain the so-called Persistence Diagram. 4. We compute the distances between the Persistence Diagrams (prior discretization of the Persistence Diagram so that it can be computed) of the different networks. 5. Finally, we analyze the similarity between different neural networks trained for the same task, for a similar task, and for a completely different task, independently of the concrete architecture, to see whether there is topological similarity.

As baselines, we set two standard matrix comparison methods that are the 1-Norm and the Frobenius norm. Having adjacency matrix $A$ and $B$, we compute the difference as $norm(A - B)$. However, these methods only work for matrices of similar size and thus, they are not general enough. We could also have used the Fast Approximate Quadratic assignment algorithm suggested in Vogelstein et al. (2015), but for large networks this method becomes unfeasible to compute.

### 3.1 PROPOSAL

Our method is as follows. We start by associating to a neural network a weighted directed graph that is analyzed as an abstract simplicial complex consisting on the union of points, edges, triangles, tetrahedrons and larger dimension polytopes (those are the elements referred as simplices). Abstract simplicial complexes are used in opposition to geometric simplicial complexes, generated by a point cloud embedded in the Euclidean space $\mathbb{R}^n$.

Given a trained neural network, we take the collection of neural network parameters as directed and weighted edges that join neurons, represented by graph nodes. Biases are considered as new vertices that join target neurons with an edge having a given weight. Note that, in this representation, we lose the information about the activation functions, for simplicity and to avoid representing the network as a multiplex network. Bias information could also have been ignored because we want large PH groups that characterize the network, while these connections will not change the homology group dimension of any order.

For negative edge weights, we reverse edge directions and maintain the absolute value of the weights. We discard the use of weight absolute value since neural networks are not invariant under weight sign transformations. This representation is consistent with the fact that every neuron can be replaced by a neuron from which two edges with opposite weights emerge and converge again on another neuron with opposite weights. From the point of view of homology, this would be represented as a closed cycle. We then normalize the weights of all the edges as expressed in Equation 1 where $w$ is the weight to normalize, $W$ are all the weights and $\zeta$ is an smoothing parameter that we set to 0.000001. This smoothing parameter is necessary as we want to avoid normalized weights of edges to be 0. This is because 0 implies a lack of connection.

$$max(1 - \frac{|w|}{max(|W|)}, \zeta) \tag{1}$$

Given this weighted directed graph, we then define a directed flag complex associated to it. Topology of this directed flag complex can be studied using homology groups $H_n$. In this work we calculate homology groups up to degree 3 ($H_0$-$H_3$) due to computational complexity and our neural network representation method's layer connectivity limit.

The dimensions of these homology groups are known as Betti numbers. The $i$-th Betti number is the number of $i$-dimensional voids in the simplicial complex ($\beta_0$ gives the number of connected components of the simplicial complex, $\beta_1$ gives the number of non reducible loops and so on). For a

---

[4]For more details, see Section 3.2.

deeper introduction to algebraic topology and computational topology, we refer to Edelsbrunner & Harer (2009); Ghrist (2014).

We work with a family of simplicial complexes, $K_\varepsilon$, for a range of values of $\varepsilon \in \mathbb{R}$ so that the complex at step $\varepsilon_i$ is embedded in the complex at $\varepsilon_j$ for $i \leq j$, i.e. $K_{\varepsilon_i} \subseteq K_{\varepsilon_j}$. In our case, $\varepsilon$ is the minimum weight of included edges of our graph representation of neural networks. Filtration parameter could also be used to select active nodes thought, the library we used does not include this capability.

The nested family of simplicial complexes is called a *filtration*. We calculate a sequence of homology groups by varying the $\varepsilon$ parameter, obtaining a persistence homology diagram. PH calculations are performed on $\mathbb{Z}_2$.

This filtration gives a collection of contained directed weighted graph or simplicial complex $K_{\varepsilon_{min}} \subseteq \ldots \subseteq K_{\varepsilon_t} \subseteq K_{\varepsilon_{t+1}} \subseteq \ldots \subseteq K_{\varepsilon_{max}}$, where $t \in [0,1]$ and $\varepsilon_{min} = 0$, $\varepsilon_{max} = 1$ (recall that edge weights are normalized).

Given a filtration, one can look at the birth, when a homology class appears, and death, the time when the homology class disappears. The PH treats the birth and the death of these homological features in $K_\varepsilon$ for different $\varepsilon$ values. Lifespan of each homological feature can be represented as an interval $(birth, death)$, of the homological feature. For each filtration, we can record all these intervals by a Persistence Barcode (PB) Carlsson (2009), or in a Persistence Diagram (PD), as a collection of multiset of intervals.

As mentioned previously, our interest in this work is to compare PDs from two different simplicial complexes. There are two distances traditionally used to compare PDs, Wasserstein distance and Bottleneck distance. Their stability with respect to perturbations on PDs has been object of different studies Chazal et al. (2012); Cohen-Steiner et al. (2005). As shown in comparative studies such as in Fasy et al. (2020), different distances and different ways of vectorising persistence diagrams have results with different levels of stability and quality.

In order to make computations feasible and to obviate noisy intervals, we filter the PDs by limiting the minimum PD interval size. We do so by setting a minimum threshold $\eta = 0.01$. Intervals with a lifespan under this value are not considered. Additionally, for computing distances, we need to remove infinity values. As we are only interested in the deaths until the maximum weight value, we replace all the infinity values by 1.0.

Wasserstein distance calculations are computationally hard for large PDs (each PD of our NN models has a million persistence intervals per diagram). Therefore we use a vectorized version of PDs instead, also called PD discretization. This vectorized version summaries have been proposed and used on recent literature Adams et al. (2017); Berry et al. (2020); Bubenik (2015); Lawson et al. (2019); Rieck et al. (2019a). For the persistence diagram distance calculation, we use the Giotto-TDA library Tauzin et al. (2020) and compute the following supported vectorized persistence summaries: 1. Persistence landscape. 2. Weighted silhouette. 3. Heat vectorizations.

## 3.2 EXPERIMENTAL FRAMEWORK

To determine the topological structural properties of trained NNs, we select different kinds of datasets. We opt for four well-known benchmarks in the machine learning community and one regarding language identification: (1) the MNIST[5] dataset for classifying handwritten digit images, (2) the Fashion MNIST Xiao et al. (2017) dataset for classifying clothing images into 10 categories, (3) the CIFAR-10[6] (CIFAR) dataset for classifying 10 different objects, (4) the Reuters dataset for classifying news into 46 topics, and (5) the Language Identification Wikipedia dataset[7] for identifying 7 different languages. For CIFAR-10 and Fashion MNIST, we pre-train a Convolutional NN (CNN), and the convolutional layers are shared between all the models of the same dataset as a feature extractor. Recall that we are focusing on MLPs, so we do not consider that convolutional weights. For MNIST, Reuters and Language Identification, we use an MLP. For Reuters and Language identification datasets, we vectorize the sentences with character frequency.

---

[5]http://yann.lecun.com/exdb/mnist/
[6]https://www.cs.toronto.edu/~kriz/cifar.html
[7]https://www.floydhub.com/floydhub/datasets/language-identification/1/data

| Number | Experiment | Index |
|---:|---|---:|
| 1 | Layer size | 1-4 |
| 2 | Number of layers | 5-9 |
| 3 | Input order | 10-14 |
| 4 | Number of labels | 15-19 |

Table 1: Indices of the experiments of the distance matrices.

We study the following variables (hyperparameters): 1. Layer width, 2. Number of layers, 3. Input order[8] 4. Number of labels (number of considered classes). We define the *base* architecture as the one with a layer width of 512, 2 layers, the original features order, and considering all the classes (10 in the case of MNIST, Fashion MNIST and CIFAR, 46 in the case of Reuters and 7 in the case of the language identification task). Then, doing one change at a time, keeping the rest of the base architecture hyperparameters, we experiment with architectures with the following configurations: 1. **Layer width**: 128, 256, 512 (*base*) and 1024. 2. **Number of layers**: 2 (*base*), 4, 6, 8 and 10. 3. **Input order**: 5 different randomizations (with *base* structure), the control experiment. 4. **Number of labels** (MNIST, Fashion MNIST, CIFAR-10): 2, 4, 6, 8 and 10 (*base*). 5. **Number of labels** (Reuters): 2, 6, 12, 23 and 46 (*base*). 6. **Number of labels** (Language Identification): 2, 3, 4, 6 and 7 (*base*). Note that this is *not* a grid search over all the combinations. We always modify one hyperparameter at a time, and keep the rest of them as in the base architecture. In other words, we experiment with all the combinations such that only one of the hyperparameters is set to a non-base value at a time. For each dataset, we train 5 times (each with a different random weight initialization) each of these neural network configurations. Then, we compute the topological distances (persistence landscape, weighted silhouette, heat) among the different architectures. In total, we obtain $5 \times 5 \times 3$ distance matrices (5 datasets, 5 random initializations, 3 distance measures). Finally, we average the 5 random initializations, such that we get $5 \times 3$ matrices, one for each distance on each dataset. All the matrices have dimensions $19 \times 19$, since 19 is the number of experiments for each dataset (corresponding to the total the number of architectural configurations mentioned above). Note that the base architecture appears 8 times (1, on the number of neurons per layer, 1 on the number of layers, 1 on the number of labels and the 5 randomizations of weight initializations). All experiments were executed in a machine with 2 NVIDIA V100 of 32GB, 2 Intel(R) Xeon(R) Platinum 8176 CPU @ 2.10GHz, and of 1.5TB RAM, for a total of around 3 days.

Note that this work focuses on MLPs, however, the method proposed is also applicable to CNNs, as they have a MLP equivalent.[9] The resulting NN is a highly sparse simplicial complex with a large number of parameters which makes PH calculation out of our computation capability for all combination of proposed experiments.

## 4 RESULTS & DISCUSSION

Results from control experiments can be seen in the third group on Figures 1 and 4. In these figures, groups are separated visually using white dashed lines. Experiments groups are specified in Table 1. Control experiments in all the images appear very dimmed, which means that they are very similar, as expected. Recall that the control experiments consist of 5 (randomizations) $\times$ 5 (executions) and that 25 different neural networks have been trained; each one of the network has more than 690,000 parameters that have been randomly initialized. After the training, results show that these networks have very close topological distance, as expected.

For Figure 2 we computed both 1-norm and Frobenius norm (the baselines) for graphs' adjacency matrices of control experiments. Note that as we ran the experiment five times, we make the mean for each value of the matrix. In order to show whether the resulting values are positive or negative, we subtract to the maximum difference of each dataset the norm of each cell separately, we take the absolute value and we divide by the maximum difference of each dataset. Therefore, we obtain five values per dataset. Table 2 shows the statistics reflecting that the distance among the experiments are large and, thus, they are not characterizing any similarity but rather an important dissimilarity.

---

[8]Order of the input features, the control experiment.
[9]https://aul12.me/machinelearning/2019/06/20/cnn-mlp-2.html

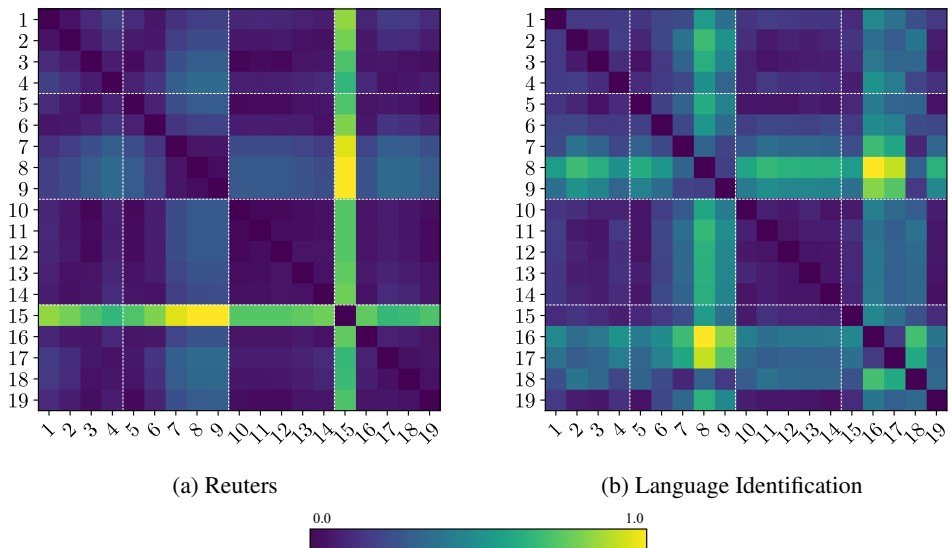

Figure 1: Distance matrices using Silhouette discretization.

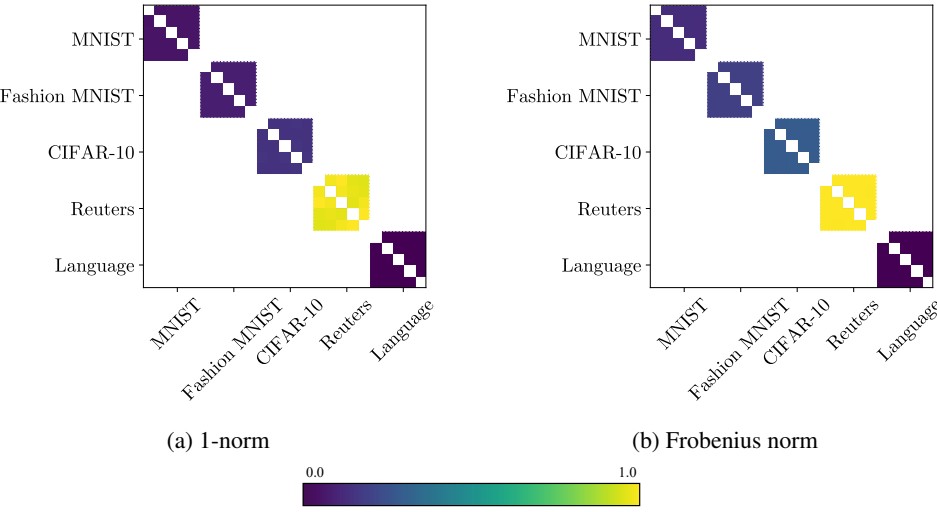

Figure 2: Control experiments using norms.

| Norm | Minimum | Maximum | Mean | Standard deviation |
|------|---------|---------|------|--------------------|
| 1-Norm | 0.6683 | 4.9159 | 1.9733 | 1.5693 |
| Frobenius | 0.0670 | 0.9886 | 0.4514 | 0.3074 |

Table 2: Normalized difference comparison of self-norm against the maximum mean distance of the experiment.

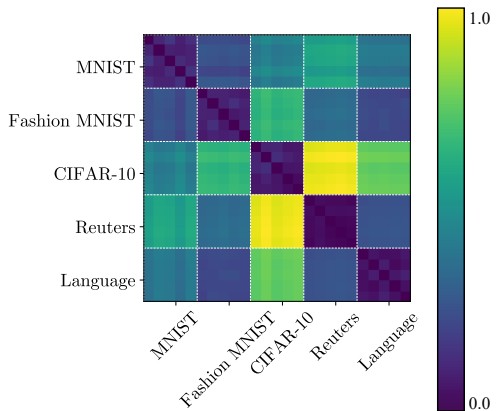

Figure 3: Control experiment comparison matrix using Silhouette discretization.

| | Heat distance | | Silhouette distance | |
|---|---|---|---|---|
| Dataset | Mean | Deviation | Mean | Deviation |
| MNIST | 0.0291 | 0.0100 | 0.1115 | 0.0364 |
| F. MNIST | 0.0308 | 0.0132 | 0.0824 | 0.0353 |
| CIFAR-10 | 0.0243 | 0.0068 | 0.0769 | 0.0204 |
| Language I. | 0.0159 | 0.0040 | 0.0699 | 0.0159 |
| Reuters | 0.0166 | 0.0051 | 0.0387 | 0.0112 |

Table 3: PH distances across input order (control) experiments, normalized by dataset.

In contrast, Figure 3, with our method (Silhouette), shows perfect diagonal of similarity blocks. In the corresponding numeric results, we obtained small distances, as shown in Table 3. We can appreciate that each dataset has its own hub. This confirms the validity of our proposed similarity measure.

The method we present also seems to capture some parts of hyperparameter setup. For instance, in Figure 4 we can observe gradual increase of distances in the first group regarding layer size meaning that, as layer size increases, the topological distance increases too. Similarly, for the number of layers (second group) and number of labels (fourth group) the same situation holds. Note that in Fashion MNIST and CIFAR-10, the distances are dimmer because we are not dealing with the weights of the CNNs. Recall that the CNN acts as a frozen extractor and are pretrained for all runs (with the same weights), such that the MLP layers themselves are the only potential source of dissimilarity between runs. Thus, our characterization is sensitive to the architecture (e.g., if we increase the capacity, distances vary), but at the same time, as we saw before, it is not dataset-agnostic, meaning that it also captures whether two neural networks are learning the same problem or not.

In Figure 4, Fashion MNIST (Figure 4b) and CIFAR (Figure 4c) dataset results are interestingly different from those of MNIST (Figure 4a) dataset. This is, presumably, because both Fashion MNIST and CIFAR use a pretrained CNN for the problem. Thus, we must analyze the results taking into account this perspective. The first fully connected layer size is important as it can avoid a bottleneck from the previous CNN output. Some works in the literature show that adding multiple fully connected layers does not necessarily enhance the prediction capability of CNNs Basha et al. (2019), which is congruent with our results when adding fully connected layers (experiments 5 to 9) that result in dimmer matrices than the one from. Concerning the experiments on input order, there is slightly more homogeneity than in MNIST, again showing that the order of sample has negligible influence. Moreover, there could have been even more homogeneity taking into account that the fully connected network reduced its variance thanks to the frozen weights of the CNN. This also supports the fact that the CNN is the main feature extractor of the network. As in MNIST results, CIFAR results show that the topological properties are, indeed, a mapping of the practical properties of neural networks. We refer to the Supplementary Material for all distance matrices for all datasets and all distances, as well as for the standard deviations matrices and experiment group statistics.

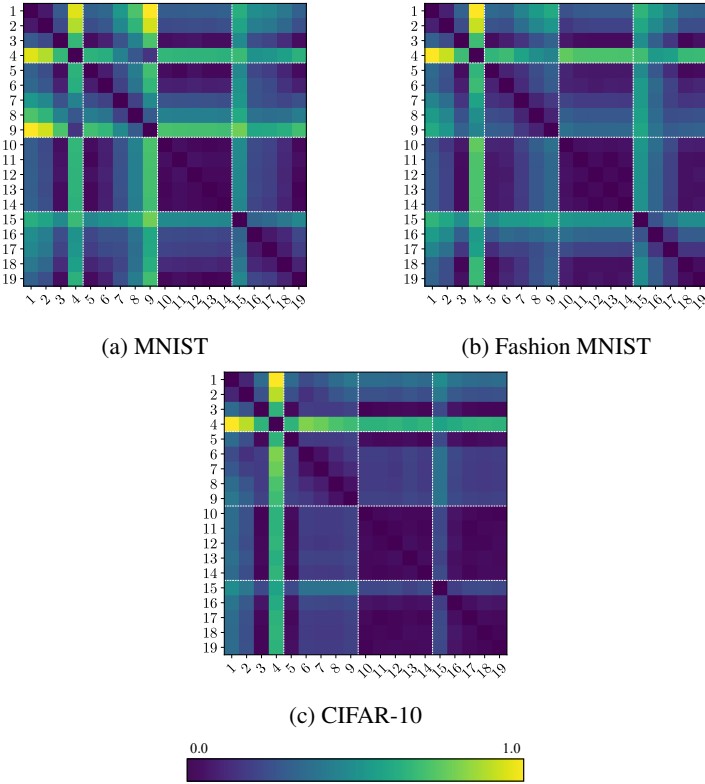

(a) MNIST

(b) Fashion MNIST

(c) CIFAR-10

Figure 4: Distance matrices using Heat discretization.

## 5 CONCLUSIONS & FUTURE WORK

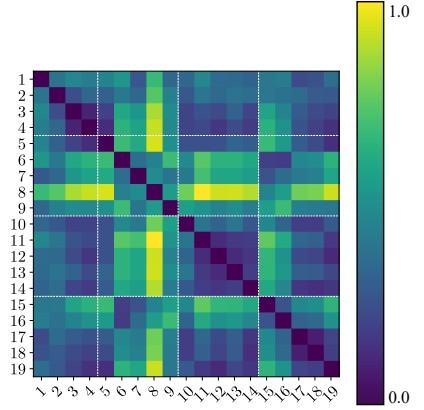

Figure 5: Language Identification (Landscape).

Results in five different datasets from computer vision and natural language lead to similar topological properties and are interpretable, which yields to general applicability. The best discretizations found are Heat and Silhouette. They show better separation of experiment groups, and are effectively reflecting changes in a sensitive way (unlike Landscape discretization).

The most remarkable conclusion comes from the control experiments. The corresponding neural networks, with different input order but the same architecture, are very close to each other. The PH framework does, indeed, abstract away the specific weight values, and captures latent information from the networks, allowing comparisons to be based on the function they approximate. The selected neural network representation is reliable and complete, and yields coherent and meaningful results. Instead, the baseline measures, the 1-Norm and the Frobenius norm, implied an important dissimilarity between the experiments in the control experiments, meaning that they did not capture the fact that these neural networks were very similar in terms of the solved problem.

Our proposed characterization does, indeed, capture meaningful properties. To the best of our knowledge, our proposed similarity measure between neural networks is the first of its kind. As future work, we suggest adapting the method to architectures such as CNNs, RNNs, and Transformers (Vaswani et al., 2017). Finally, we suggest performing more analysis regarding the learning of a neural network from a topological point of view.

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
