# OpenReview forum: "Characterizing and Measuring the Similarity of Neural Networks with Persistent Homology "
_ICLR.cc/2022/Conference — ICLR 2022 Submitted_

### Official Review · Reviewer_WhHV · 2021-10-31

**Correctness:** 3
**Technical Novelty And Significance:** 3
**Empirical Novelty And Significance:** 3
**Recommendation:** 5
**Confidence:** 3

**Main Review:**

## Strength
1.	This paper presents a new approach to represent neural networks which includes abstract simplicial complex representation and PH method. It is novel and insightful to combine neural networks similarity measure and topological methods. Especially, using PD discretization to measure the similarity is intuitive and reasonable.
2.	This paper is well-supported theoretically. Both the topological characterizations associated to neural networks and PH methods for measuring similarity are technically sound and significant for the problem studied.
3.	This paper is well written and easy to understand. Although I think the display of experimental results could be arranged more reasonably (see weaknesses 1)

## Weakness
1.	As mentioned in the introduction, a correct characterization should be able to distinguish whether two neural networks are trained for the same task, regardless of subtle differences in architecture, input order, etc. However, discussion about how PD discretization recognizes the different types of tasks better than 1-norm and Frobenius norm (similar to figure3), not only in the control experiments, is not clear and sufficient.
2.	Very few details are given about the calculation of PD discretization, i.e., Persistence landscape, Weighted silhouette and Heat vectorizations. Their calculation and topological meaning may explain partly why Heat and Silhouette separate the experimental group better than Landscape, which is worth exploring as well.



**Summary Of The Paper:**

In order to characterize and measure the similarity of two neural networks, this paper uses abstract simplicial complex to represent neural networks. The Persistent Homology of the constructed graph associated to a neural network is computed to obtain the corresponding Persistence Diagram. To calculate and quantify the similarity, the authors use supported vectorized persistence summaries: Persistence landscape, Weighted silhouette, and Heat vectorizations respectively and compare their ability to measure the similarities between neural networks. Detailed and complete experiments are conducted on different kinds of datasets. Each experiment only contains one modification on hyperparameter.  Results show that the PH-based representation did characterize and capture latent information from the networks.

**Summary Of The Review:**

The motivation and the problem studied in this paper is interesting. The authors present a novel and effective approach to represent neural networks in a topological way and propose a similarity measure using PD discretization, which is supported by the experimental results. However, the discussion part could be more focused on the discretization’s ability to separate the experiment groups, rather than sensitivity to parameter changes.

---

> ### Author Response · Authors · 2021-11-20
> **Response**
>
> > As mentioned in the introduction, a correct characterization should be able to distinguish whether two neural networks are trained for the same task, regardless of subtle differences in architecture, input order, etc. However, discussion about how PD discretization recognizes the different types of tasks better than 1-norm and Frobenius norm (similar to figure3), not only in the control experiments, is not clear and sufficient.
>
> The main problem we have encountered in including a comparison table with the 1-norm and Frobenius norm between adjacency matrices is that the comparison is only possible when the matrices have the same dimensions. We compare NNs with very different architectures. Perhaps it could be applied in the case of control examples (different initializations of the NN weights and changes in the order of the examples during training) to measure the distances between the same architectures.
>
> > Very few details are given about the calculation of PD discretization, i.e., Persistence landscape, Weighted silhouette and Heat vectorizations. Their calculation and topological meaning may explain partly why Heat and Silhouette separate the experimental group better than Landscape, which is worth exploring as well.
>
> You can find more about stability in the paper we cite in the related work section (https://openreview.net/forum?id=X1bxKJo5_qL). However, Heat and Silhouette by definition take into account more structure than the Landscape. We could add something about this in the supplementary material.
>
> > ​​The motivation and the problem studied in this paper is interesting. The authors present a novel and effective approach to represent neural networks in a topological way and propose a similarity measure using PD discretization, which is supported by the experimental results. However, the discussion part could be more focused on the discretization’s ability to separate the experiment groups, rather than sensitivity to parameter changes.
>
> Diagrams are discretized to make distance computations faster as the Wasserstein distance is extremely hard to compute. We can add more insight about the discretization methods and why some of them are better than others, however, the paper we mentioned in the previous paragraph is a very good resource to understand this point. With respect to the “separate the experiment groups” we might not have understood it correctly but we artificially grouped the experiments depending on the change that we were making to the NN so that they would be easier to see the differences and the changes.
>
>
> Thanks for your time, effort and comments.

---

### Official Review · Reviewer_HwgX · 2021-11-01

**Correctness:** 3
**Technical Novelty And Significance:** 2
**Empirical Novelty And Significance:** 2
**Recommendation:** 5
**Confidence:** 4

**Main Review:**

Strength
- This paper shows that the distance between graphs calculation using TDA is more suitable for comparing tasks than the general method of calculating the distance between graphs.
- Constructed a framework for comparing trained NNs and demonstrated its effectiveness for simple methods.

Weakness
- As mentioned in the Related work section of this paper, Rieck et.al.(2019b) have shown a method for representing NNs as graphs and analyzing them with TDA. On the other hand, when the intrinsic graph structures of NNs are similar, they correspond to almost the same task, and when the tasks are completely different, the NN structures are often very different.（It is possible that the structure of NNs will be similar for different tasks, but I don’t think we need to worry about it practically.）The proposed method is based on the fact that NNs for similar tasks have similar graph structures, and TDA is good at analyzing graph structures. Therefore, the idea of the proposed method is not particularly new.
- The structure of a trained NN depends on the training data, loss function, and optimization algorithm. Even if the NN is for the same task, the structure of the NN is often different if they are different. In this study, verification under different conditions is lacking, and we cannot say that the proposed method measures whether the NNs are for the same task or not. However, it is worthwhile as a comparison under the same conditions, so it will be a sufficient contribution if specific applications are shown.
Minor comment
Table captions are usually written above the table.


**Summary Of The Paper:**

In this paper, the authors give a method to evaluate the closeness of a task considered by a neural network. They represent a trained NN as a weighted graph and extract features of the NN by calculating persistent homology from the graph.
The distance between NNs is calculated by calculating the distance between persistent homologies, and it is experimentally shown that the corresponding tasks of each NN can be determined whether they are similar or different.

**Summary Of The Review:**

The contribution to the proposal and validation of a more effective framework for comparing trained NNs is acknowledged. On the other hand, the novelty of the idea is low and its applicability is questionable.

[Post rebuttal comment] I appreciate the author's responses. The author's perception was clear to me, but I think it still needs to be improved with additional validation, so I did not change my grade.

---

> ### Author Response · Authors · 2021-11-20
> **Response**
>
> > As mentioned in the Related work section of this paper, Rieck et.al.(2019b) have shown a method for representing NNs as graphs and analyzing them with TDA. On the other hand, when the intrinsic graph structures of NNs are similar, they correspond to almost the same task, and when the tasks are completely different, the NN structures are often very different.（It is possible that the structure of NNs will be similar for different tasks, but I don’t think we need to worry about it practically.）The proposed method is based on the fact that NNs for similar tasks have similar graph structures, and TDA is good at analyzing graph structures. Therefore, the idea of the proposed method is not particularly new.
>
> Rieck et.al.(2019b) don't analyze complete graph topology; they study consecutive pairs of layers. They study only the H_0 group, i.e. connectivity. The global topology of the network is not taken into account (there are trivial global transformations of an NN that are not captured by analysing pairs of layers).
> Also absolute values for edge weights are used. In our case, edge direction is important (for that reason we use flag complexes). Rieck representation can’t be used for comparing different NNs.
>
> (we will be adding an additional comment on this matter)
>
> > The proposed method is based on the fact that NNs for similar tasks have similar graph structures, and TDA is good at analyzing graph structures. Therefore, the idea of the proposed method is not particularly new.
>
> The proposed method is based on the fact that NNs for similar tasks have similar homological complexity, measured by persistence diagram distance. Different experiments with a distinct number of layers and layer densities are proposed. Associated graphs are very different but homological properties are maintained.
>
> > The contribution to the proposal and validation of a more effective framework for comparing trained NNs is acknowledged. On the other hand, the novelty of the idea is low and its applicability is questionable.
>
> The paper develops a representation associated with the NNs that can be compared for different architectures trained on very different problems. This representation is independent of initialisations or rearrangements of the examples on which it has been trained. The main application is the characterisation and comparison of NNs in model selection, analysis of their learning characteristics and generalisation.
>
> On the applicability of our method, please see our concurrent paper showing that we can monitor the learning process (https://openreview.net/pdf?id=TNxKD3z_tPZ). The method that we propose in this work can be used to monitor the learning of NNs, to stop their training early, to select better architectures and to understand the impact of the hyperparameters and tweak them, among others.
>
> Many thanks for your comments and dedication.

---

> > ### Comment · Reviewer_HwgX · 2021-11-21
> > **Discussion**
> >
> > Thank you for your clarification. But, there seems to be a misunderstanding of my point. The story of your proposal should flow as follows:
> >
> > 1. The intrinsic structure of NNs as graphs of similar tasks is similar.
> > 2. So, if we can calculate the similarity as a graph of NNs, we can regard it as the similarity of NNs in terms of tasks.
> > 3. TDA can be used to calculate the similarity of 2 because of its high performance in analyzing graph structure.(Rieck et. al. proposed to view NNs as graphs and to apply TDA to their analysis.)
> > 4. Creating an optimal TDA calculation method for calculating the similarity of NNs from a task perspective can be used as a method for calculating the similarity of NNs from a task perspective
> > This story is not explicitly stated, but I believe it is already generally recognized. It is my understanding that your suggestion relates to 4, and that your suggestion has a contribution for it, including what is in your comment.
> >
> > Regarding the last point, the paper should be self-contained and should not use other papers under submission as evidence. Also, from the point of view of double-blindness, I do not consider this comment because of the concern about knowing the author information.

---

> > > ### Author Response · Authors · 2021-11-21
> > > **Response**
> > >
> > > > 1. The intrinsic structure of NNs as graphs of similar tasks is similar.
> > >
> > > We understood the following:
> > > “The intrinsic structure of NNs, as graphs, of similar tasks is similar.”
> > >
> > > The intrinsic structure of NNs, as topological objects (Persistent Homology objects specifically) associated to the NN, of similar tasks is similar.
> > > We analyze the topological properties of graphs by employing persistent homology.
> > >
> > > > 2. So, if we can calculate the similarity as a graph of NNs, we can regard it as the similarity of NNs in terms of tasks.
> > >
> > > We have graph1 of NN1 and graph2 of NN2 (regardless of the architecture), we compute PH for both (this results in a Persistence Diagram), we then compute the similarity of the Persistence Diagrams with three different discretization methods.
> > >
> > > > 3. TDA can be used to calculate the similarity of 2 because of its high performance in analyzing graph structure. (Rieck et. al. proposed to view NNs as graphs and to apply TDA to their analysis.)
> > >
> > > Rieck’s method is applied on pairs of layers and it cannot be used to compare two different neural networks. Additionally, their method only applies to the H_0 homology group. This means that the method is only taking into account connectivity (yes/no). Our method, depending on the architecture, can take into account as many homology group orders as needed.
> > >
> > > To sum up:
> > > 1. When consecutive layer analysis is used, the global topology of the network is not taken into account (there are trivial global transformations of an NN that are not captured by analysing pairs of layers).
> > > 2. There are trivial modifications of an NN, adding pairs of neurons with reversed weight axes, which create an equivalent NN but the above representations fail.
> > > 3. Corneanu et al. and Rieck et al. use absolute value of weights. Suppose reversing the sign of all weights in a trained NN, is it the same NN? Absolutely no.
> > >
> > >
> > > > 4. Creating an optimal TDA calculation method for calculating the similarity of NNs from a task perspective can be used as a method for calculating the similarity of NNs from a task perspective. This story is not explicitly stated, but I believe it is already generally recognized. It is my understanding that your suggestion relates to 4, and that your suggestion has a contribution for it, including what is in your comment.
> > >
> > > This paper proposes and validates a method on how to represent the NNs as topological objects that can be compared. The benefits of our representation method are the following:
> > >
> > > 1. It is invariant under training sample reordering.
> > > 2. It is invariant under weight initialization.
> > > 3. We can compare NNs of any architecture (number of neurons per layer, number of layers). Even though NN1 and NN2 have different architecture.
> > >
> > > > Regarding the last point, the paper should be self-contained and should not use other papers under submission as evidence. Also, from the point of view of double-blindness, I do not consider this comment because of the concern about knowing the author information.
> > >
> > > We agree with the necessity of the paper to be self-contained that is why we did not add any reference to the paper. We think the paper is self-contained, as we are exploring how to compute the similarity of the networks. You then asked for other applications, and as this paper is self-contained in the way we explore the similarity, we did not explore other usages. Thus, we have been forced to provide as an additional resource the other paper.

---

> > > > ### Author Response · Authors · 2021-11-22
> > > > **More clarifications/discussion on the importance of our method**
> > > >
> > > > We have found the following problems in existing topological models in the literature (mainly Rieck et al. and Corneanu et al.).
> > > >
> > > > ## (1) Invariance to superfluous cycle insertions
> > > >
> > > > Suppose that given a NN we perform the following transformation: We add two neurons and connect their input to a single neuron in a lower layer and their two outputs to a single neuron in an upper layer. If we assign opposite weights to the edges connecting these new neurons to the lower layer (e.g., a and -a) and similarly assign opposite weights to the edges connecting them to the upper layer (e.g., b and -b), we obtain another equivalent neural network. Since the weights are opposite, the net weight added on the upper layer neuron is zero and therefore both NNs generate the same outputs for any input.
> > > >
> > > > We expect a topological model representing the NN to be invariant to this kind of equivalence operation; the addition of superfluous cycles (I call them cycles because if we take sense of their edges according to their sign, as we do in our model, it is a homologically non-significant cycle, for any degree homology group).
> > > >
> > > > However, in Rieck et al. neural persistence model, since the a and b values can be as high or as low as we want, it is possible to cancel them out, relatively speaking, the weights of the rest of the axes that are subsequently used in the neural persistence calculation.
> > > >
> > > > ## (2) Invariance to identity layer insertions
> > > > Suppose we add an identity layer. That is, for any given NN layer, we duplicate its neurons and trivially connect them with edges whose weight is maximum.
> > > >
> > > > The neural network obtained after this transformation is functionally equivalent to the original NN. In our persistent homology model, as these edges have the maximum weight, the transformed value of the weight is zero, i.e. the duplicate nodes are always connected. From our point of view of persistent homology, no homology group of any degree is altered.
> > > >
> > > > From the layer pair point of view of Rieck neural persistence, four 0-homology elements are generated (since H_0 only evaluates connectivity), as many as the number of neurons in the duplicated layer exists.
> > > >
> > > >
> > > > ## (3) Non-planar neural networks
> > > > Sometimes we want to represent neural networks that are not planar, i.e. that connect several networks (like siamese NNs), for example by creating an assembly between them, or because they are networks that use multiple connections between non-consecutive layers (as in the example of residual connections). In these cases, the Rieck et al. model is not applicable because it is only valid for planar NNs with contiguous layers.
> > > >
> > > >
> > > > ## (4) Models based on correlation between activations of neurons
> > > > Finally, we have found models based on correlations of the neuron activations (Corneanu et al.). They form an undirected weighted graph that represents these correlations (it is an undirected graph since it does not take into account that the correlation can be negative). The persistent homology analysis is then performed on this graph. We have not found this type of model useful as they depend on the input data and not only on the function under study. We wanted to study neural networks as functions over the entire input domain, regardless of the data used for training.
> > > >
> > > > We do not think data-based approach is positive for these reasons (among others):
> > > > 1. The need to pass data through the network. How much data?
> > > > 2. It is not the same to pass {training, validation, testing} data.
> > > > 3. The subset may not be representative.
> > > > 4. Topology/Structure of network is not taken into account. It is not being captured.
> > > > 5. Activations may not be enough.
> > > >
> > > > ## Final thoughts
> > > >
> > > > In general, we think that in order to make a complete homology representation of a NN we should use higher order homology groups (we have found important groups up to degree 3 but there could be cases where the complexity is higher). This is necessary to find non-trivial higher order homology structures that are specific and characterise the NN. We also see it as very important to employ directed graphs, and therefore to use directed flag homology.

---

### Official Review · Reviewer_rtBj · 2021-11-03

**Correctness:** 2
**Technical Novelty And Significance:** 2
**Empirical Novelty And Significance:** 2
**Recommendation:** 3
**Confidence:** 4

**Main Review:**

The article is well written and proposes an interesting approach, but I have some trouble figuring out how powerful and meaningful the method actually is, due to the lack of theoretical back-ups and the vague, hand-waving interpretations of the set of experiments.

I have the following comments:

a. I understand that thresholding persistence diagrams is important for keeping the running time reasonable, but, on the other hand, setting the lifespan threshold to 0.01 seems pretty arbitrary. It would be nice to comment about how this threshold was chosen in the text.

b. It is quite difficult to interpret the proposed distance matrices and values. Increasing and decreasing values of distances can be caused by many factors, and not only the ones suggested by the authors. For instance, simply increasing the size of the graphs sometimes results in larger distances just because the persistence diagrams have more points, invalidating some of the authors interpretation. In order to be fully convincing, I think some null statistical model should be provided and compared to, with some corresponding p-values. See for instance https://openreview.net/pdf?id=rHaiOtGdRS. Otherwise, it is impossible to go beyond vague comments and explanations of the results.

c. I am not sure about how useful are these distances. Even though they seem to detect a few properties of the network, they do not seem to always be very discriminative: some architectures with very different hyper parameters coming from different experiment groups can end up quite close in the proposed distance. Indeed, values that are outside of the blocks corresponding to the various experiments can still be very small in the distance matrices. Hence, I am not sure if there are settings in which using this distance actually make sense, practically speaking. Would it be possible, for instance, to do model selection based on it?

d. Topological uncertainty (https://www.ijcai.org/proceedings/2021/367), is a good reference to add in the related work section, since it also aims at characterizing the shape of neural nets with persistence (for another application though).

[Post rebuttal comment] Even though I appreciate the author's responses and suggestions, I still think that the paper requires substantial improvements before publication, so I did not change my grade.


**Summary Of The Paper:**

In this work, the authors propose to characterize neural networks with Topological Data Analysis, more precisely with its main descriptor, the so-called persistence diagram, in order to be able to compare neural networks with different numbers of layers, different numbers of neurons, or trained on different data sets. More specifically, they show that the computational graphs corresponding to the neural networks can be filtered using the edge weights that are learnt during training, in order to produce filtered flag complexes, from which persistent homology can be computed. Then, the authors interpreted the distances between the persistence diagrams obtained from networks with varying parameters (number of layers, neurons, labels), and showed that, often, the distances have intuitive correlation with the complexity of the networks.


**Summary Of The Review:**

While the approach is novel and interesting, I think the experiments are not convincing enough and too hand-waving to definitely validate the procedure. Since the approach is purely experimental, I think the work is too preliminary for publication.

---

> ### Author Response · Authors · 2021-11-20
> **Response I**
>
> > The article is well written and proposes an interesting approach, but I have some trouble figuring out how powerful and meaningful the method actually is, due to the lack of theoretical back-ups and the vague, hand-waving interpretations of the set of experiments.
>
> You are right on the vague definition of the similarity among problems in which NNs are trained and compared with others. We need a similarity measure across problems approximated with NNs. This measure should be compared with the results of our topological distance. We believe that a similarity measure could be based on the Shannon’s entropy/information measure.
>
> > a. I understand that thresholding persistence diagrams is important for keeping the running time reasonable, but, on the other hand, setting the lifespan threshold to 0.01 seems pretty arbitrary. It would be nice to comment about how this threshold was chosen in the text.
>
> This threshold is limited mainly because of our computation capacity. However, we assessed the impact of the threshold and with the one we selected the results were that very small noisy triplets were filtered out and were not contributing with information of any kind. Regarding the discretizations there is a very interesting paper (https://openreview.net/forum?id=X1bxKJo5_qL) that we were based on to guide our decisions.
>
> > b. It is quite difficult to interpret the proposed distance matrices and values. Increasing and decreasing values of distances can be caused by many factors, and not only the ones suggested by the authors. For instance, simply increasing the size of the graphs sometimes results in larger distances just because the persistence diagrams have more points, invalidating some of the authors interpretation.
>
> Homology groups do not have a cardinality associated with the number of nodes and edges of the source graph. For example, the homology groups associated with any triangularisation of a torus are the same regardless of the number of triangles used. We want to use homology as a tool precisely because of this ability to abstract from the concrete realisation of the metric object (the weighted graph associated with the NN).
>
> > In order to be fully convincing, I think some null statistical model should be provided and compared to, with some corresponding p-values. See for instance https://openreview.net/pdf?id=rHaiOtGdRS. Otherwise, it is impossible to go beyond vague comments and explanations of the results.
>
> We can include the statistical significance tests. However, note that the results are constant and that there is a considerable similarity, thus, there won’t be any change. Also note that 5 runs by 5 randomization control experiments is extremely . We understand that this might be necessary for some researchers to further validate the results and we will be including it as well as the raw results.
>
> > c. I am not sure about how useful are these distances. Even though they seem to detect a few properties of the network, they do not seem to always be very discriminative: some architectures with very different hyper parameters coming from different experiment groups can end up quite close in the proposed distance. Indeed, values that are outside of the blocks corresponding to the various experiments can still be very small in the distance matrices.
>
> First of all we would like to assure the positive results of the control experiments. Then, we would like to note that the architecture changes in the experiments are performed separately. It was not a grid search. Therefore, there can be experiments of other groups that have the same architecture and thus are similar. For instance: Figure 4(a): 19 with respect to 3, 5 and control experiments is similar as they are all using the base distance, so they are equivalent to the control experiments. We recomputed those even though they are the same for (1) extending more the control experiments over the base architectures (2) show them in the matrix ordered.
>
> > Hence, I am not sure if there are settings in which using this distance actually make sense, practically speaking. Would it be possible, for instance, to do model selection based on it?
>
> It is the first iteration of many more. We studied that PH can be associated with problems solved by different NNs. The representation that we came up with is invariant with respect to different initializations and reordering of training samples. In a posterior paper we have studied the learning evolution showing that we can monitor the learning process (concurrent paper on ICLR https://openreview.net/pdf?id=TNxKD3z_tPZ). The method that we propose in this work can be used to monitor the learning of NNs, to stop their training early, to select better architectures and to understand the impact of the hyperparameters and tweak them, among others.

---

> > ### Comment · Reviewer_rtBj · 2021-11-22
> > **Response**
> >
> > Thank you for the time taken to answer my comments. I have updated my review.
> >
> > > Homology groups do not have a cardinality associated with the number of nodes and edges of the source graph. For example, the homology groups associated with any triangularisation of a torus are the same regardless of the number of triangles used. We want to use homology as a tool precisely because of this ability to abstract from the concrete realisation of the metric object (the weighted graph associated with the NN).
> >
> > I agree, this is why I said "sometimes". My comment was more about the fact that generally speaking, on simplicial complexes with generic, random values on the simplices, the size of the PD will be correlated to the number of simplices. Of course, this is not always the case, for instance, as you mentioned, when dealing with triangulations. However, this motivates the comparison and statistical tests against a null model of a completely random complex, to ensure that the proposed distance is significantly different than artifacts arising from the complex size.

---

> ### Author Response · Authors · 2021-11-20
> **Response II**
>
>
> > d. Topological uncertainty (https://www.ijcai.org/proceedings/2021/367), is a good reference to add in the related work section, since it also aims at characterizing the shape of neural nets with persistence (for another application though).
>
> Thank you for the reference. We were not aware of this work given the time proximity to our own paper. The orientation of our paper is different. We are not interested in studying activations (like Corneanu et al) because it depends on input data. We are interested in studying trainned NNs. Our goal is to compare NNs. The representation proposed in the Topological Uncertainty paper is similar to the work from Corneanu et al that we already cited.
>
> Thanks for your time and dedication.

---

### Official Review · Reviewer_ZGgm · 2021-11-04

**Correctness:** 1
**Technical Novelty And Significance:** 2
**Empirical Novelty And Significance:** 3
**Recommendation:** 5
**Confidence:** 5

**Main Review:**

I think the paper is interesting and novel. The idea of converting the NN that way to PD is something I have not seen before. However, I have many concerns about the paper.

Neural networks are functions, and it it is not clear if two neural networks that have different architectures should be not similar. This method suffers from this in my opinion. You might have two neural networks that are totally different, one of them is giant and one of them is small and yet they are similar in the function they do--this will not be captured by the the method presented--at least this is not justified.

- also, more importantly, why do you want to measure similarity between two NNs? I think it is interesting mathematical purpose but it is not clear from a practical perspective why you want or need to do that?

**Summary Of The Paper:**

The paper converts a given NN to a weighted directed graph and consider the flag complex on the top of it and use that object to compute the PD of the input NN. Two neural networks are then considered to be similar iff the PDs are close enough with respect to WD.

**Summary Of The Review:**

While the paper is interesting and offers fresh point of view, I feel the paper lacks justification of important parts : such as why do we need a similarity between two NNs.

---

> ### Author Response · Authors · 2021-11-20
> **Response**
>
> > I think the paper is interesting and novel. The idea of converting the NN that way to PD is something I have not seen before. However, I have many concerns about the paper.
>
> Actually we convert the NN to a sequence of simplicial complexes (associated to the weights of the NN). Over this sequence, we compute the PD that we compare.
>
> > Neural networks are functions, and it it is not clear if two neural networks that have different architectures should be not similar. This method suffers from this in my opinion. You might have two neural networks that are totally different, one of them is giant and one of them is small and yet they are similar in the function they do--this will not be captured by the the method presented--at least this is not justified.
>
> Most of the existing methods to compare NN do not take into account the problem you mentioned. For instance, the existing methods cannot compare adjacency matrices of different sizes. However PH of two simplicial complexes of different sizes are comparable. In fact, in our experiments we observe that NNs of different sizes are more similar than NNs trained for different problems.
>
> > also, more importantly, why do you want to measure similarity between two NNs? I think it is interesting mathematical purpose but it is not clear from a practical perspective why you want or need to do that?
>
> The numeric characterization of the NNs (in terms of weights and bias) would allow it to control its learning and generalization capabilities. In this paper we analyze the existence of an additional characterization using PH and the distance among PDs, which can be applicable in (1) monitoring the learning process (2) understanding the impact of the hyperparameters (3) finding equivalent architectures and therefore improving the Neural Architecture Search processes.
>
> > While the paper is interesting and offers fresh point of view, I feel the paper lacks justification of important parts : such as why do we need a similarity between two NNs.
>
> Before this paper there was no way to represent a NN that would allow to compare it independently to the dimensions of the NNs. We think that this paper unlocks a future work in the terms explained in the previous paragraph.
>
> Thanks for your time and dedication.

---

> > ### Comment · Reviewer_ZGgm · 2021-11-27
> > **the paper is still in a poor shape and non of my comments have been addressed**
> >
> > I do not see the authors did any effort to address my comments. I still think the paper is a reject and my comments above still hold--the paper has many claims that are not justified.

---

> > > ### Author Response · Authors · 2021-11-30
> > > **Response**
> > >
> > > Dear ZGgm reviewer.
> > >
> > > We tried to address your comments. Could you please copy the comments that have not been addressed? alternatively you could reply to our attempt to address your comments.
> > >
> > > We tried to do our best. Thanks for your understanding.

---

### Public Comment · ~Yuri_Smirnov1 · 2021-11-15
**Missing citation**

A missing citation. Since you employ the persistence diagrams of complexes (PD), here is the reference where they were first introduced, under the name of canonical forms : Barannikov, S. "The Framed Morse Complex and its Invariants", Advances in Soviet Mathematics, 21: 93–115 (1994). Also, the computation of persistence diagrams is based on the algorithm described in section 2.1 of this reference.

---

> ### Author Response · Authors · 2021-11-20
> **Response**
>
> Dear Yuri,
>
> We hope you enjoyed reading our paper. You are right on the missing citation.
>
> Thanks for your time and dedication.

---

### Decision · Program_Chairs · 2022-01-20

**Decision:**

Reject

**Comment:**

*Summary:* Compare neural networks and tasks using TDA, particularly persistence diagrams.

*Strengths:*
- Some reviewers found this a fresh perspective.
- Distance calculation using TDA can offer advantages and a theoretical basis.

*Weaknesses:*
- Insufficient motivation and experimental evidence for utility of the proposed approach.
- Computational cost and hyperparameter choices in PD computation.
- Difficulty of interpreting proposed distance matrices.

*Discussion:*

ZGgm found the paper interesting and that it offered a fresh perspective, but that the purpose of the comparison was not sufficiently well motivated. The authors provide some explanations, particularly about the method allowing to compare networks of different sizes, but ZGgm found their comments were not adequately addressed. rtBj found that even though the authors made efforts to address their comments, the paper still requires substantial improvements. HwgX appreciated the authors’ responses but considers that the paper needs to be improved with additional validation. They expressed doubts about the adequacy of the approach and found that although it improves upon certain methods, it is insufficiently verified.


*Conclusion:*

All reviewers agree that this work has some strengths but also significant weaknesses and does not reach the acceptance bar for this conference. Main weaknesses are insufficient motivation and experimental evidence. The reviewers made several suggestions on how the paper could be improved. I agree with the reviewers and hence I must reject this article.